# Rice bran oil emulgel as a pork back fat alternate for semi-dried fish sausage

**Manat Chaijan**[1], **Ling-Zhi Cheong**[2], **Worawan Panpipat** [1]*

**1** Department of Food Science and Innovation, Food Technology and Innovation Research Center of Excellence, School of Agricultural Technology and Food Industry, Walailak University, Nakhon Si Thammarat, Thailand, **2** Department of Food Science and Engineering, School of Marine Science, Ningbo University, Ningbo, China

* pworawan@wu.ac.th

## Abstract

The objective of this study was to investigate the effects of rice bran oil emulsion filled gels (EG) substitution for pork back fat on the characteristics of Chinese style semi-dried tilapia sausage (CFS). EG prepared using different gelling agents and processing conditions were used as pork back fat alternate in the CFS. From the results, physical, chemical and micro-structural qualities of CFS were governed by the type of EG incorporated. Regarding the overall quality, CFS formulated with carrageenan-EG was classified as an optimal formula. CFS added with carrageenan-EG showed a superior oxidative stability, color preservation, and water holding capacity compared to the control during vacuum packaged storage at room temperature for 20 days. Thiobarbituric acid reactive substances (TBARS) and microbial quality of both formulae remained in the acceptable level (TBARS < 1 mg/kg and total plate count < 4 log CFU/g) throughout the storage. Therefore, the carrageenan based EG substitution for pork back fat is a promising avenue for the production of the CFS where a high saturated animal fat was substituted by vegetable oil.

## Introduction

Sustainability and healthy alternate choices of foods are the future of agricultural production. High saturated fat intake has become restrict due to its adverse health effects particularly increasing the risk of cardiovascular diseases (CVD) [1]. One of the highest foods with rich saturated fatty acids is sausage, which is typically formulated with 20–30% animal fats [2]. Animal fats are also a source of cholesterol which obviously increases the risk factors of CVD [3, 4]. Several attempts have been done to partially or completely replace the animal fats by the vegetable oils for sausages [5–7]. Among oil used for this purpose, rice bran oil (RBO) is one of the possible candidates due to the health benefits. RBO has been recommend to help reduce the risk of lipid disorders and related symptoms [8]. RBO contains about 44% oleic acid, 30% linoleic acid, 1% linolenic acid, 2% stearic acid, and 23% palmitic acid [9]. The ratio of saturated, monounsaturated and polyunsaturated fatty acids (PUFA) at 1:1.5:1 was found in RBO which was similar to the value proposed by the World Health Organization and American Heart Association for lowering blood cholesterol level [10]. RBO also consists of numerous bioactive

**Data Availability Statement:** All relevant data are within the manuscript.

**Funding:** This research was financially supported by the new strategic research project (P2P),

Walailak University, Thailand. The funders had no role in study design, data collection and analysis, decision to publish, or preparation of the manuscript.

**Competing interests:** The authors have declared that no competing interests exist.

antioxidant compounds including tocopherols, γ-oryzanol, and tocotrienols, leading to its oxidative stability and health benefits [11]. Hence, RBO can be applied as a potential animal fat alternate to produce the healthier sausage.

Chinese-style fish sausage (CFS) or *Pla Chiang* (in Thai) is a semi-dried coarse sweet sausage produced by hand kneading coarse ground fish and coarse ground animal fat with salt, nitrite, spices, and sugar. After stuffing in the edible casing, the drying is performed to lower the moisture content to 25% for extending the shelf life [12]. CFS is needed to be cooked e.g. pan frying before consumption. This kind of sausage is thus characterized as non-emulsion typed sausage because an emulsion-gel structure is not formed. This is a challenge to substitute the vegetable oil for animal fat in this type of sausage to produce healthier product. It is also difficult to achieve similar product appearance, rheology, and sensory properties because the chemical and physical characteristics of lard and vegetable oils are different. Typically, saturated fats or solid fats from animal sources are necessary to retain the sausage shape at room temperature [12]. To tackle this technological problem, oil filled in the gel matrix based emulsion delivery systems namely "emulgel (EG)" or "emulsion filled gel" as a fat alternate have been prepared [13].

EG is the soft semi-solid materials which is structurally incorporated emulsion droplets into a polymeric gel matrix [14]. EG could be a better option in comminuted emulsion like sausage than simple emulsified oil to achieve a better physical characteristic close to animal fat with greater physical stability because the product stability can greatly be improved by immobilizing oil droplets within a three dimensional gel network, resulting in the prevention of flocculation and coalescence [14]. A better texture, higher water holding capacity, and lower cooking loss of products containing EG can be obtained compared to simple emulsified oil [15]. Gelling agents from proteins (milk, soy, gelatin, and egg) and hydrocolloids (carrageenan, starch, and gum) are typically used to encourage EG formation [14–18]. The gelation conditions such as temperature, pH, and enzyme treatment also affect to the EG characteristics [16]. Thus, the characteristics of EG are highly altered by gelling agent and gelling process.

Here, protein based and hydrocolloid based gelling agents with different gelling processes were comparatively employed to produce RBO based EG. The effects of EG substitution for pork back fat on the characteristics of CFS were consequently investigated. A CFS made with pork back fat was used as a control. The substitution of pork back fat by EG may improve the perception of CFS in the Halal and Kosher markets.

## Material and methods

### Food additives

Carrageenan, whey protein isolate (WPI), sodium caseinate (SC), konjac flour were food grade and purchased from Chemephan Co., Ltd. (Bangkok, Thailand). King® RBO was purchased from Thai Edible Oil Co., Ltd. (Bangkok, Thailand).

### Preparation of emulgel (EG)

Four types of EG were prepared using different gelling agents (carrageenan, WPI, SC, and konjac flour). Carrageenan based EG was produced using the method of Poyato et al. [15] with some modification. The oil phase was prepared by pre-heating the mixture of RBO (40 g) and polysorbate 80 (0.12 g) at 70˚C for 2 min. Then, the oil phase was homogenized with 59.88 g of preheated 1.5% aqueous κ-carrageenan suspension (70˚C/2 min) at 16,000 rpm for 3 min (IKA® homogenizer, Model T25 digital Ultra-Turrax®, Staufen, Germany). After cooling down to room temperature (28-30˚C), sample was incubated at 4˚C for 12 h before being used. WPI based EG was produced using the method of Oliver et al. [17] with some

modification. To prepare the pre-emulsion, preheated RBO (70°C/2 min) was homogenized with preheated 3% (w/w) aqueous WPI dispersion (70°C/2 h/gently stirring occasionally with a glass rod) at a ratio of 1:1 (w/w) at 9000 rpm for 3 min. Then, the pre-emulsion was homogenized with preheated 9% (w/w) aqueous WPI solution (70°C/2 h) at a ratio of 1:1 (w/w) at 9000 rpm for 3 min. Consequently, glucono-delta-lactone (GDL) was added to obtain the final concentration of 0.63% w/w of the aqueous phase for developing an EG. The mixture was stored at 25°C for 20 h and subsequently kept at 4°C for 12 h. SC based EG was produced using the method of Oliver et al. [17] with some modification. To prepare the pre-emulsion, preheated RBO (70°C/2 min) was homogenized with preheated 5% (w/w) aqueous SC dispersion (70°C/2 h/gently stirring) at a ratio of 1:1 (w/w) at 9000 rpm for 3 min. Then, the pre-emulsion was homogenized with 14.5% (w/w) aqueous micellar casein isolate (MCI) solution at a ratio of 1:1 (w/w) at 9000 rpm for 3 min. Thereafter, GDL was added to obtain the final concentration of 2.87% (w/w) of the aqueous phase. The mixture was incubated at 25°C for 20 h to proceed the gelation and the obtained EG was stored at 4°C for 12 h. Konjac based EG was produced using the method of Jiménez-Colmenero et al. [18] with some modification. From the technical data sheet, the konjac flour composed of carbohydrate (75%), moisture (12%), protein (8%), fat (<1%), and ash (4%). The oil phase was prepared by pre-heating the mixture of RBO (100 g) and polysorbate 80 (0.3 g) at 70°C for 2 min. The water phase was made by homogenizing konjac flour (45 g) and κ-carrageenan (9 g) in warm distilled water (583 mL) at 9,000 rpm for 8 min. The oil phase was gradually added into the water phase under a continuous homogenization (9000 rpm for 3 min) to obtain the emulsion. Pre-gelatinized corn starch powder (27 g) was dispersed in 145.7 mL distilled water and further homogenized with the emulsion at 9,000 rpm for 3 min. The mixture was cooled down to 10°C and then 90 mL of 1% calcium hydroxide was added. The mixture was gently mixed at room temperature until the EG was obtained. The obtained EG was kept at 4°C for 12 h before being used

## Production of CFS formulated with EG

CFS were produced according to the method of Panpipat and Chaijan [12]. The lipid content of all formulae was equally adjusted to 6.28%, according to the lipid content in the commercial CFS in Thailand. To obtain a similar fat level, each EG was calculated based on lipid content and used to replace the pork back fat. Briefly, tilapia mince (62.8 g) was handed mixed with lard or EG (to obtain 6.28% fat), sugar (15.7 g), sodium nitrite (1.73 g), and sodium erythorbate (0.15 g) for 8 min. The handed kneading was continued for 2 min after adding 3.92 g of iced water. Then, the batters were stuffed into cellulose casings (2.5 cm in diameter, B.O.T Co., Ltd., Bangkok, Thailand) and the CFS were hand tied with cotton strings at 6 cm intervals with 50 g each sausage link. After drying at 60°C for 3 h in a tray drier (Owner Foods Machinery Co., Ltd., Bangkok, Thailand), the CFS with ~25% moisture content were vacuum packed (Multivac A300/16, Sepp Haggmüller KG, Wolfertschwenden, Germany) and stored at room temperature overnight before measurement.

## Physicochemical properties of CFS made with and without EG

**Proximate composition.** The proximate composition of CFS including moisture, protein, fat, ash, and fiber was analyzed according to the method of AOAC [19]. Carbohydrate content was estimated by difference (1).

$$\text{Carbohydrate content } (\%) = 100 - [\text{moisture} + \text{protein} + \text{fat} + \text{ash} + \text{fiber}] \tag{1}$$

**pH.** Ground samples (2 g) were homogenized with distilled water (10 mL) for 2 min under ice bath. The pH of homogenates were measured using a Cyberscan 500 pH meter (Eutech, Singapore).

**Yield.** Yields were analyzed after the manufacturing process of CFS. The yield was calculated from the following Eq (2):

$$Yield\ (\%) = \left(\frac{Weight\ after\ processing}{Weight\ of\ the\ raw\ material}\right) \times 100 \tag{2}$$

**Total expressible drip.** The total expressible drip was calculated based on the percentage of the sample weight before and after being compressed by 5-kg standard weight for 2 min where a thin sliced sample (5-mm thickness) was placed between pieces of Whatman No. 1 filter paper [20].

**Color analysis.** Color values of CFS, including $L^*$, $a^*$, and $b^*$, were measured by a portable Hunterlab Miniscan/EX instrument (10˚standard observers, illuminant D65, Hunter Assoc. Laboratory; VA, USA).

**Lipid oxidation.** Lipid oxidation indices including peroxide value (PV) and thiobarbituric acid reactive substances (TBARS) were analyzed according to Chaijan et al. [21] and Buege and Aust [22], respectively. For the PV, lipid (1 g) previously extracted by the method of Folch et al. [23] was mixed with 25 mL of a chloroform:acetic acid mixture (2:3, v/v). Then, 1 mL of saturated potassium iodide (KI) was added. After keeping in the dark for 5 min, 75 mL of distilled water and 0.5 mL of 1% starch solution were added. The PV was estimated by titrating the iodine liberated from KI with a standardized 0.01 M sodium thiosulfate and expressed as milliequivalents of free iodine (meq)/kg.

For the TBARS, a ground sample (0.5 g) was homogenized with 2.5 mL of a mixed solution (0.375% (w/v) thiobarbituric acid, 15% (w/v) trichloroacetic acid, and 0.25 M hydrochloric acid). The mixture was heated in a boiling water bath (95-100˚C) for 10 min. After cooling down with running tap water, the centrifugation was applied at 3600 ×g at 25˚C for 20 min (RC-5B plus, Sorvall, Norwalk, CT, USA) and the absorbance of the supernatant was measured at 532 nm. A standard curve was prepared using 1,1,3,3-tetramethoxypropane (0 to 10 ppm) and the TBARS content was reported as mg malondialdehyde (MDA) equivalent/kg.

**Textural analysis.** Three pieces of CFS with the thickness of 2.5 cm were subjected to texture profile analysis (TPA). Hardness, springiness, cohesiveness, adhesiveness, gumminess, and chewiness values were obtained from 2-cycle compression at a speed of 127 mm/min using a TPA compression plate equipped with a TA-XT2 texture analyzer (Stable Microsystem, UK).

**Morphological analysis.** The scanning electron microscope (SEM) (GeminiSEM, Carl Ziess Microscopy, Germany) was used for microstructure analysis of CFS [24]. CFS with a thickness of 2–3 mm were fixed with 2.5% (v/v) glutaraldehyde in 0.2 M phosphate buffer (pH 7.2) for 3 h. Thereafter, samples were rinsed for 10 min in distilled water and subsequently dehydrated in ethanol with serial concentrations of 25, 50, 70, 80, 90, and 100% (v/v). Samples were critical point dried using $CO_2$ as transition fluid. Dried samples were mounted on a bronze stub, sputter-coated with gold and analyzed using an SEM at an acceleration voltage of 10 kV.

**Changes in quality of CFS formulated with and without EG during storage.** The selected CFS formulated with EG and the control (100% pork back fat) were vacuum packaged and stored at room temperature for 20 days. Every 5 days interval, color, hardness, total expressible drip, PV, TBARS, and pH were analyzed as mentioned above. Water activity ($a_w$) was measured using an AquaLab CX-2 (Decagon Devices Inc., Pullman, WA, USA). Free fatty

acid (FFA) content was measured according to the procedure of Lowry and Tinsley [25]. The total viable count (TVC) and yeast and mold counts were determined according to the American Public Health Association's method [26].

**Statistical analysis.** A completely randomized design was used in this study and the entire experiment was replicated three times. The triplicate determinations were done for all analyses. Data analysis was processed by the Statistical Package for the Social Sciences (SPSS) 10.0 for Windows (SPSS Inc., Chicago, IL, USA). Duncan's multiple-range test was used to identify significant differences (p<0.05) among samples. For pairwise comparison, the t-test was used.

## Results and discussion

### Proximate composition

The proximate compositions of CFS formulated with and without EG are shown in Table 1. There were no statistically significant differences in the moisture and fat contents of CFS formulated with and without EG (p>0.05), since the drying was performed until the target moisture content of about 25% was obtained. For the fat, similar contents in all batters were initially fixed at 6.28%. After drying, the fat content of all CFS were roughly raised to 12% (Table 1). For protein, fiber, ash and carbohydrate contents, some differences among CFS were observed. This was related to the EG composition. The incorporation of protein based EG (WPI and SC) obviously altered the protein content of the corresponding CFS (p<0.05), ranging between 22.01% and 23.60% (Table 1). The highest protein content in SC-EG based CFS (p<0.05) was due to the inclusion of SC in the EG preparation. However, carrageenan-EG, konjac-EG, and lard-based CFS showed slight differences in protein contents (19.04–19.44%). CFS formulated with both protein based EG and the control sample showed the highest ash content, followed by carrageenan-EG, and konjac-EG formulated CFS (p<0.05). The

**Table 1. Proximate composition and physicochemical characteristics of semi-dried Chinese style tilapia sausages formulated with and without different types of emulgel (EG).**

| Parameters | Carrageenan based EG | Whey protein isolate based EG | Sodium caseinate based EG | Konjac based EG | Pork back fat (control) |
|---|---|---|---|---|---|
| Proximate composition | | | | | |
| Moisture (%) | 25.06±0.52a | 25.51±0.13a | 25.33±0.32a | 25.43±0.11a | 25.42±0.12a |
| Protein (%) | 19.04±0.04c | 22.01±0.03d | 23.60±0.28e | 19.22±0.16b | 19.44±0.03a |
| Fat (%) | 12.86±1.20a | 12.72±1.34a | 12.64±0.11a | 12.53±0.06a | 12.28±0.41a |
| Ash (%) | 4.70±1.65b | 5.10±1.58c | 5.85±1.70c | 3.90±0.06a | 5.72±0.11c |
| Fiber (%) | 5.74±0.36c | 5.25±0.12b | 4.20±0.08a | 5.42±0.22b | 5.53±0.42bc |
| Carbohydrate (%) | 32.60±1.49c | 29.41±6.94a | 28.38±8.79a | 33.50±0.34c | 31.61±0.63b |
| pH | 6.27±0.02b | 6.27±0.02b | 5.50±0.08a | 6.33±0.07b | 6.24±0.08b |
| Yield (%) | 93.51±2.15a | 90.74±2.12a | 93.60±1.17a | 91.95±1.77a | 91.97±0.77a |
| Total expressible drip (%) | 4.74±0.53b | 4.79±0.57b | 10.82±0.70c | 4.43±0.60b | 2.99±0.09a |
| Color | | | | | |
| $L^*$ | 15.99±0.48a | 16.30±0.61ab | 17.92±0.52c | 17.98±0.50c | 17.16±0.23bc |
| $a^*$ | 1.99±0.23a | 2.50±0.07b | 2.11±0.16ab | 2.32±0.23ab | 2.01±0.30a |
| $b^*$ | 1.80±0.27b | 2.98±0.24c | 1.68±0.08b | 2.95±0.31c | 1.17±0.15a |
| Peroxide value (PV) (meq/kg) | 1.03±0.02a | 1.78±0.43c | 1.82±0.34c | 1.07±0.04a | 1.29±0.09b |
| Thiobarbituric acid reactive substances (TBARS) (mg malondialdehyde equivalent/kg) | 0.34±0.02a | 0.83±0.87c | 0.87±0.64c | 0.39±0.19a | 0.61±0.78b |

Values are given as mean±standard deviation (SD) from triplicate determinations. Different letters in the same row indicate significant differences (p<0.05).

lowest ash content in the konjac-EG formulated CFS was possibly due to a low content of ash in the konjac flour.

Similarly, a variation in fiber content was observed among formulations in which the highest fiber content was found in carrageenan-EG made CFS, followed by the control sample, konjac-EG ≈ WPI-EG, and CS-EG made CFS ($p < 0.05$). This was possibly due to different fiber contents in the ingredients. All formulations contained high carbohydrate content, ranging from 28.38 to 33.50% (Table 1). This was owing to the high amount of sugar added in this type of sausage. In addition, the uses of carbohydrate based EG, particularly carrageenan and konjac flour, could be an extra-source of carbohydrate. From the results, the difference in proximate composition in each formulation was governed by the composition and content of EG employed.

## pH, yield, and total expressible drip

The pH values of all CFS were in the range of 5.50–6.33 (Table 1). No significant differences in pH were observed in all samples ($p > 0.05$) except for SC-EG made CFS. A lower pH of this EG was due to the addition of a chemical acidulant, GDL, to induce the gel formation. GDL slowly converted to gluconic acid in the presence of water, resulting in a pH reduction [27]. The yields of CFS made with lard and various EG are given in Table 1. No significant differences in yield was observed among samples ($p > 0.05$). Thus, the type of EG had a comparable effect on processing yield of CFS. This was in line with the study of Jiménez-Colmenero et al. [24] who reported no significant changes in yield of frankfurter replaced pork back fat by emulsified olive oil. Mittal and Blaisdell [28] suggested that the proportion of protein/fat ratio in sausage highly influenced the product weight loss during thermal processing. Here, there were no differences in fish protein/fat ratios among formulations and thus a comparable yield was obtained. Total expressible drips of CFS made with and without EG are displayed in Table 1. Higher exudate was found in all EGs made CFS compared to the control ($p < 0.05$). Results implied that the proteins did not effectively bind water in the sausage matrix when EG was added. The highest total expressible drip was found in CS-EG made CFS with the pH of 5.5 (Table 1). At a pH of 5.5, which was closer to the muscle protein isoelectric point, the water binding capacity was minimal. Overall, all EG might partially destabilize the sausage structure, resulting in the imbalance among protein-protein, protein-water, and protein-fat interactions. The incorporation of lipid into hydrophilic EG may compete the binding of polypeptide chains and water, resulting in the exudation of water from the gel network.

## Color

The effects of different EGs on the color parameters of CFS are shown in Table 1. The highest lightness ($L^*$) was found in 3 formulations including SC-EG made CFS, konjac-EG made CFS, and the control CFS, followed by WPI-EG made CFS, and carrageenan-EG made CFS ($p < 0.05$). The significant differences were also found for redness value ($a^*$) in EG formulated CFS compared to the control ($p < 0.05$), except the carrageenan-EG made CFS in which a comparable $a^*$ value to the control was observed ($p > 0.05$). All EG formulated CFS showed a higher yellowness value ($b^*$) than the control ($p < 0.05$). This was due to the expression of natural color of RBO and gelling agent used for EG preparation. This reflected that the color of CFS cannot be maintained when the lard was replaced by EG. This was in agreement with the finding of Poyato et al. [15] who found the significant difference in $L^*$, $a^*$ and $b^*$ values of Bologna-type sausage made with carrageenan based linseed oil filled gel compared to the normal emulsified oil and lard formulated sausages.

## Lipid oxidation

The lowest PV and TBARS values were found in CFS added with carrageenan based EG- and konjac based EG (p<0.05), suggesting the greater oxidative stability of both formulae. This might be due to the direct antioxidant activities and filling abilities of both carrageenan and konjac flour. Sokolova et al. [29] reported the *in vitro* ferric reducing activity and scavenging activities against hydroxyl radical and superoxide anion radical. The antioxidant activity of konjac flour has been reported by Jiang et al. [30]. Interestingly, both protein based EG, WPI and SC, rendered the CFS with higher degree of lipid oxidation than the others (p<0.05). This was probably due to the presence of oxidized proteins in the original WPI and SC or the oxidation of those proteins during CFS production. The oxidized proteins may further enhance the lipid oxidation. Giblin et al. [31] reported that the content of protein carbonyls in milk and milk products significantly increased after heat treatment. However, TBARS contents of all samples were lower than the rancidity limit of 1.0 mg MDA/kg sample [32]. From the results, the incorporation of unsaturated fatty acid-containing RBO in gel matrix could be a promising means to improve the oxidative stability of CFS.

## Textural properties

Typically, the meat protein forms a gel network during heating which can hold the components therein. Thus, lipid droplet may tightly retained in the gel components, and then buried inside the meat protein gel matrix [33]. The textural properties of CFS are presented in Table 2. Hardness represents the force required to compress the sample to attain a given deformation [34]. Springiness is the elastic recovery that occurs when the compressive force is taken away [34]. Cohesiveness is a capacity in breaking down the internal structure [34]. Adhesiveness represents the work required to overcome the attractive forces between the surfaces of food and other materials [35]. Gumminess is defined as the product of hardness and cohesiveness and it is a characteristic of semisolid foods with a low degree of hardness and high degree of cohesiveness [35]. Chewiness represents the energy required for chewing until it is ready for swallowing [34]. From the results, the textural parameters of CFS were governed by EG type. EG is a biphasic system composed of emulsion gelled by gelling agents [36]. Thus, the type of gelling agents used for EG structuration can definitely influence the textural characteristics of the products therein. Although EG in this study were all soft-solid materials they may behave differently in the CFS matrix and hence influenced the textural properties of the final CFS products. CFS added with all EG had higher cohesiveness than the control (p<0.05), indicating a better consistency of the products. Poyato et al. [15] reported that the gelling agent in the EG could compensate the pore or cavity in protein network, resulting in the formation of dense, which was a more continuous and homogenous meat protein gel network. In addition, the biopolymeric interfaces may have

**Table 2. Textural properties of semi-dried Chinese style tilapia sausages formulated with and without different types of emulgel (EG).**

| Parameters | Carrageenan based EG | Whey protein isolate based EG | Sodium caseinate based EG | Konjac based EG | Pork back fat (control) |
|---|---|---|---|---|---|
| Hardness (N) | 4.96±0.46ab | 4.28±0.31a | 6.67±0.80c | 5.97±0.41bc | 5.07±0.84ab |
| Springiness (mm) | 3.77±0.43ab | 3.43±0.02a | 4.25±0.12b | 3.81±0.35ab | 3.41±0.31a |
| Cohesiveness | 0.22±0.01bc | 0.19±0.02b | 0.31±0.02d | 0.23±0.02c | 0.15±0.02a |
| Adhesiveness (N.mm) | 2.92±0.74ab | 2.64±0.72ab | 2.21±0.30a | 2.86±0.28ab | 3.58±0.24b |
| Gumminess (N) | 1.12±0.16ab | 0.83±0.12a | 2.12±0.42c | 1.38±1.13b | 0.80±0.21a |
| Chewiness (N) | 4.28±1.07ab | 2.87±0.43a | 9.03±2.05c | 5.28±0.94b | 2.76±0.96a |

*Values are given as mean±standard deviation (SD) from triplicate determinations. Different letters in the same row indicate significant differences (p<0.05).

acted as an anchor between the droplets and the gel matrix [16]. Compared to the control, CFS added with WPI-EG and carrageen-EG tended to have comparable hardness, springiness, adhesiveness, gumminess, and chewiness to the control (p>0.05). Generally, WPI provided the gel with soft solid structure and low cohesiveness [37]. Thus, WPI-EG tended to have the soft texture like pork back fat. In the case of carrageenan-EG, carrageenan can form the interaction with myofibrillar proteins to produce a viscoelastic gel [38]. Carrageenans are used extensively in meat products such as cooked hams, poultry products, and sausages due to their ability to bind water and control the texture and structural integrity [38, 39].

It should be noted that SC based EG rendered the CFS with the highest hardness and chewiness, indicating the hardening of this product. The pH of the CFS made with SC based EG was 5.5 (Table 1) which was closed to the pI of myofibrillar proteins. Myofibrillar proteins tended to be aggregated at the pI and only poor gels were formed [40]. This was in agreement with the highest expressible drip of this formula (Table 1). For CFS added with konjac-EG, the cohesiveness, gumminess, and chewiness were significantly higher than the control (p<0.05). This was possibly due to the unique hydrogel properties of konjac glucomannan [41] which can improve the viscoelastic properties of the myofibrillar protein gel matrices.

## Morphology

The appearances of EG and pork back fat as well as the SEM images of the CFS formulated with and without EG are observed in Fig 1. From the SEM images, an irregular surface was displayed in all samples. The control CFS presented a rough structure with small pores and suspended lipid beads distributed in the protein network (Fig 1E). The larger pores and lipid beads were seen in the EG incorporated samples compared to the control. This was due to the fact that the control was added with pork back fat where most of the lipid was solidified in the subcutaneous adipose tissue [42]. Thus, the lipid seemed to be uniformly distributed in the CFS. In the case of EG, they were made from rice bran oil emulsion embedded in the gel matrices. Consequently, the size distribution and the stability of the lipid droplets in the CFS were governed by the emulsion stability and gel network integrity [14–17]. From the results, CFS made with SC-EG (Fig 1C) and konjac-EG (Fig 1D) carried large lipid particles compared to other treatments. In the case of SC-EG, the agglutination of myofibrillar proteins and subsequently coalescence of lipid droplets may be taken place since the pH of the product was close to pI as mentioned above. For CFS made with konjac-EG, the lipid droplets were smaller than that with SC-EG. This was possibly due to the gel-forming ability of konjac glucomannan [41] which can delay the flocculation and/or coalescence of the oil droplets. Among EG made CFS, carrageenan-EG (Fig 1A) and WPI-EG (Fig 1B) rendered the CFS with similar microstructure to the control (Fig 1E), having homogenous lipid beads and small spongy pores. The molecular characteristics of gelling agents in the EG may account for their different relative affinities for the droplet surfaces [43]. Since the oil was emulsified into a dispersion of carrageenan or denatured WPI prior to gelation, it is reasonable to assume that an interfacial layer of these biopolymers uniformly surrounded the oil droplets [44]. The results were in line with the texture profile where CFS added with carrageenan-EG and WPI-EG tended to have comparable hardness, springiness, adhesiveness, gumminess, and chewiness to the control (Table 2). Moreover, carrageen-EG seemed to provide the CFS with a homogeneous network. Ortiz and Aguilera [45] reported that carrageenan microgels can be trapped within the gelated protein matrix and produced a homogenous composite gel.

## Changes in quality of CFS during storage

Overall, carrageenan-EG and WPI-EG substitution for pork back fat had negligible impact on chemical composition, textural and microstructural properties of CFS. However, carrageenan-

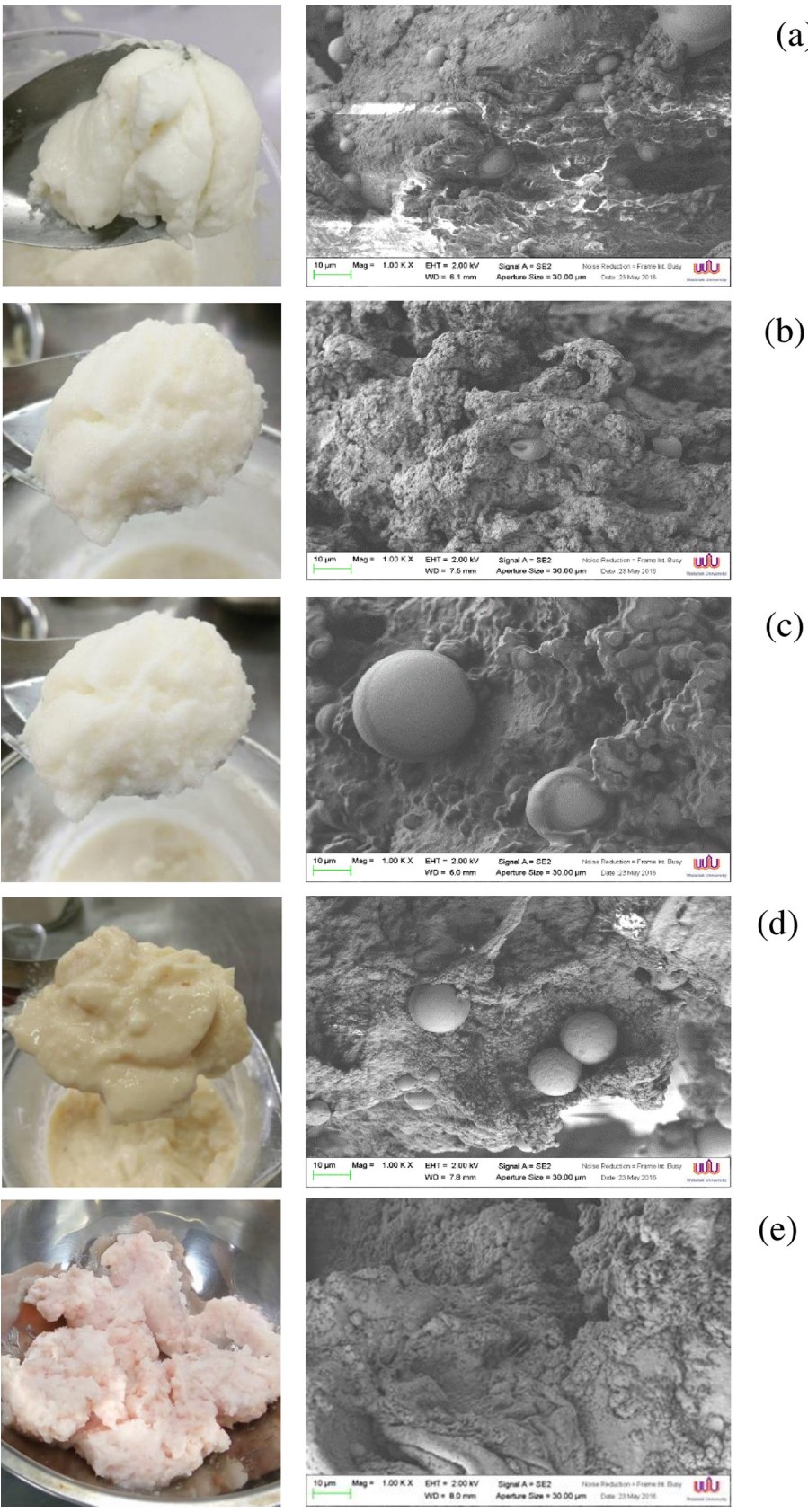

**Fig 1. Electron microscopic images of semi-dried Chinese style tilapia sausages formulated with and without different types of emulgel (EG).** (magnification: 1000× EHT: 1.0 kV). Carrageenan based EG (a), whey protein isolate based EFG (b), sodium caseinate based EG (c), konjac based EG (d), and control (pork back fat) (e).

EG rendered the CFS with a better oxidative stability as evidenced by lowered PV and TBARS content. Thus, CFS added with carrageenan-EG was selected as the optimum formula and used for the storage test in comparison with the control.

## Changes in color, hardness, and total expressible drip

Changes in color parameters of CFS with and without EG during storage are presented in Fig 2. Overall, the lightness ($L^*$) (Fig 2A), redness (+$a^*$) (Fig 2B), and yellowness (+$b^*$) (Fig 2C) of both samples increased with time (p<0.05), indicating the discoloration of all samples. An increase in lightness was mainly caused by the released water which can reflect more light whereas the increases in the yellowness and redness were probably due to the occurrence of the Maillard reaction products. At the end, the control CFS showed a higher redness and yellowness with a lower lightness than the EG formulated CFS, suggesting the darkening of the former.

The hardness of EG formulated CFS remained constant over the storage period (p>0.05), while an increasing trend in hardness was found in the control (p<0.05) (Fig 2D). Results indicated that the EG was able to maintain the hardness of the sausage during storage. An increase in hardness value of the control sample was probably due to the water loss during storage (Fig 2E). Although, the total expressible drip of both sausages tended to increase after 5 days, the remarkable increase in total expressible drip was found in the control (p<0.05) (Fig 2E). The water binding capacity of carrageenan may have helped the EG to retain more water in the CFS matrix.

## Changes in lipolysis and lipid oxidation

The FFA content of both CFS progressively increased over 20 days of storage (Fig 3A). The residual endogenous lipase/phospholipase and the microbial enzymes may cause the lipolysis in the CFS during storage. However, a lower FFA content was observed in CFS added with EG. The incorporation of lipid into carrageenan gel matrix could reduce the enzymes accessibility. Hur et al. [46] reported that the encapsulated egg yolk in chitosan or pectin had a lower rate of lipid and cholesterol digestion than non-encapsulated sample.

Lipid oxidation, one of main quality deteriorations of vegetable oil formulated meat products, was due to an increased PUFA content compared to animal fat formulation. Changes in PV and TBARS of CFS formulated with EG compared to the control are displayed in Fig 3B and 3C, respectively. The increases in PV and TBARS with storage periods were observed (p<0.05). However, TBARS contents of both samples were below the rancidity limit of 1.0 mg MDA/kg sample [31]. The results indicated that an incorporation of RBO in carrageenan gel matrix prior to use in CFS production could potentially protect the autooxidation. de Souza Paglarini et al. [47] reported that TBARS of soybean oil EG formulated Bologna sausage increased during storage but the contents were below the rancidity limits. Although lard contains higher amount of saturated fatty acids, it still contains unsaturated fatty acids which can be oxidized. In addition, the EG structure may have helped to prevent the susceptibility of lipid oxidation. In EG formulated sausage, oils might be buried in the gel matrix which can restrict the oxygen penetration. Thus, the lipid oxidation in control was higher than sample with EG.

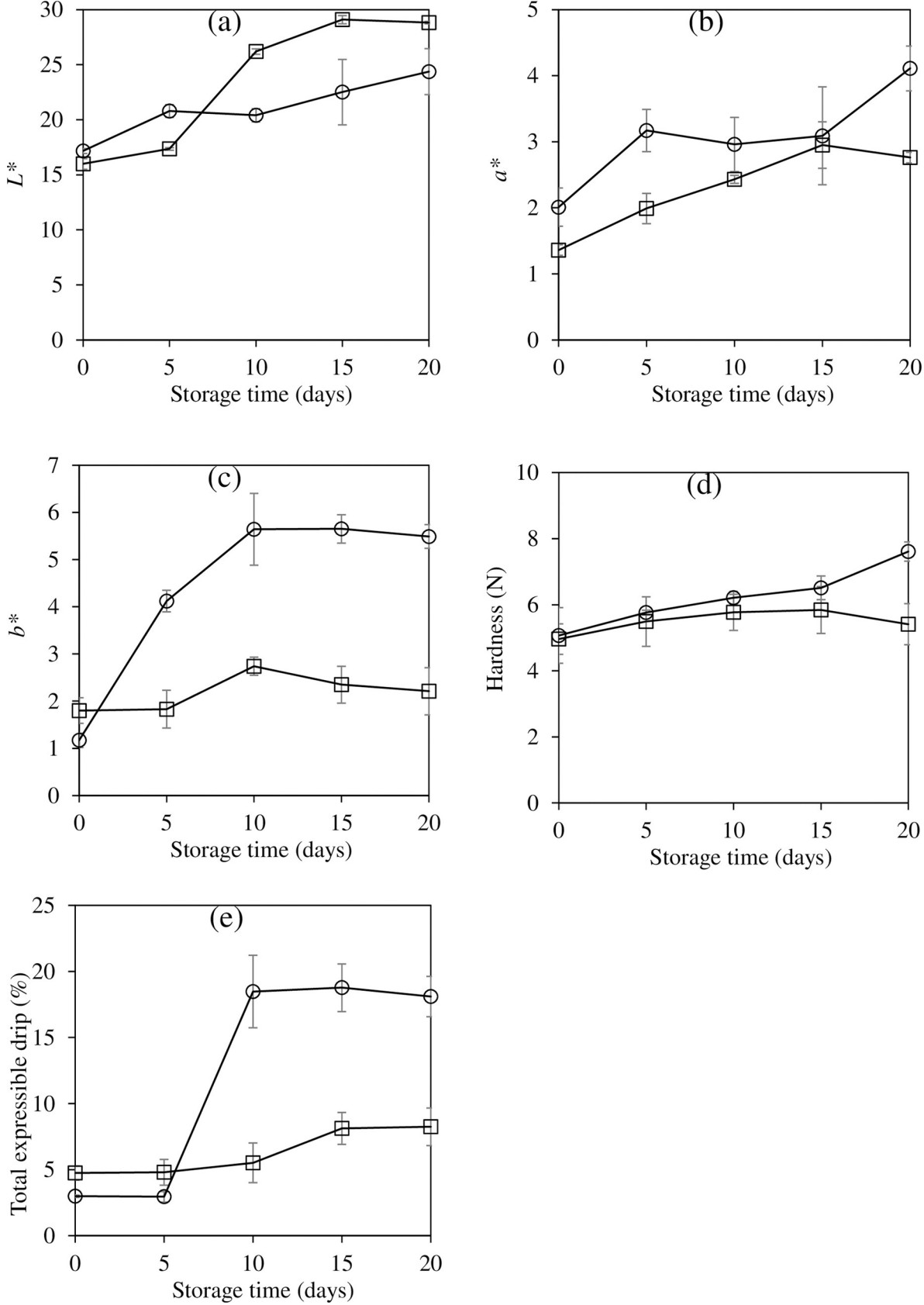

**Fig 2.** Changes in $L^*$ value (a), $a^*$ value (b), $b^*$ value (c), hardness (d), and total expressible drip (e) of semi-dried Chinese style tilapia sausages formulated with pork back fat (control; ○) and carrageenan based emulgel (□) during storage at room temperature.

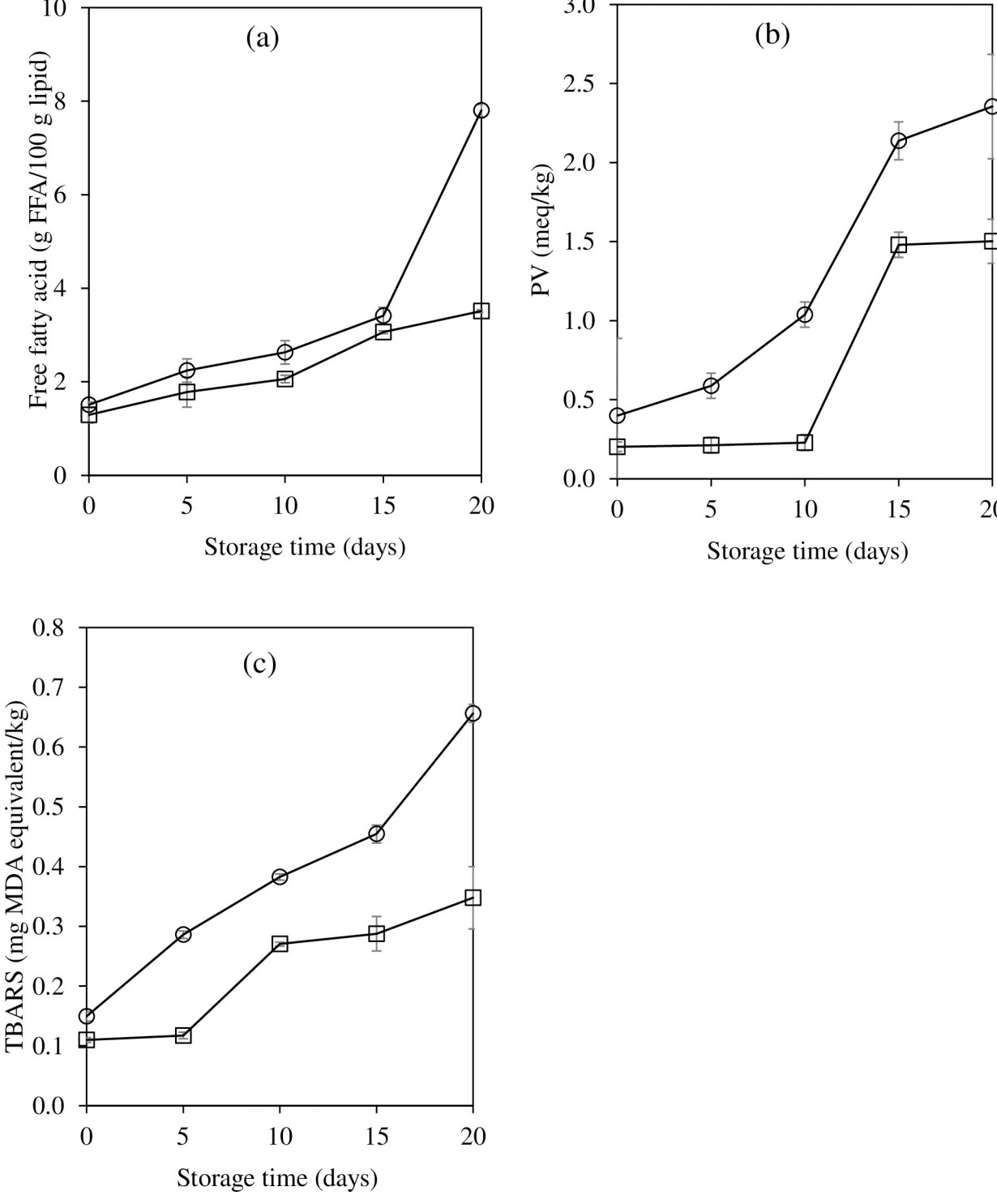

**Fig 3.** Changes in free fatty acid content (a), peroxide value (PV) (b), and thiobarbituric acid reactive substances (TBARS) content (c) of semi-dried Chinese style tilapia sausages formulated with pork back fat (control; ○) and carrageenan based emulgel (□) during storage at room temperature.

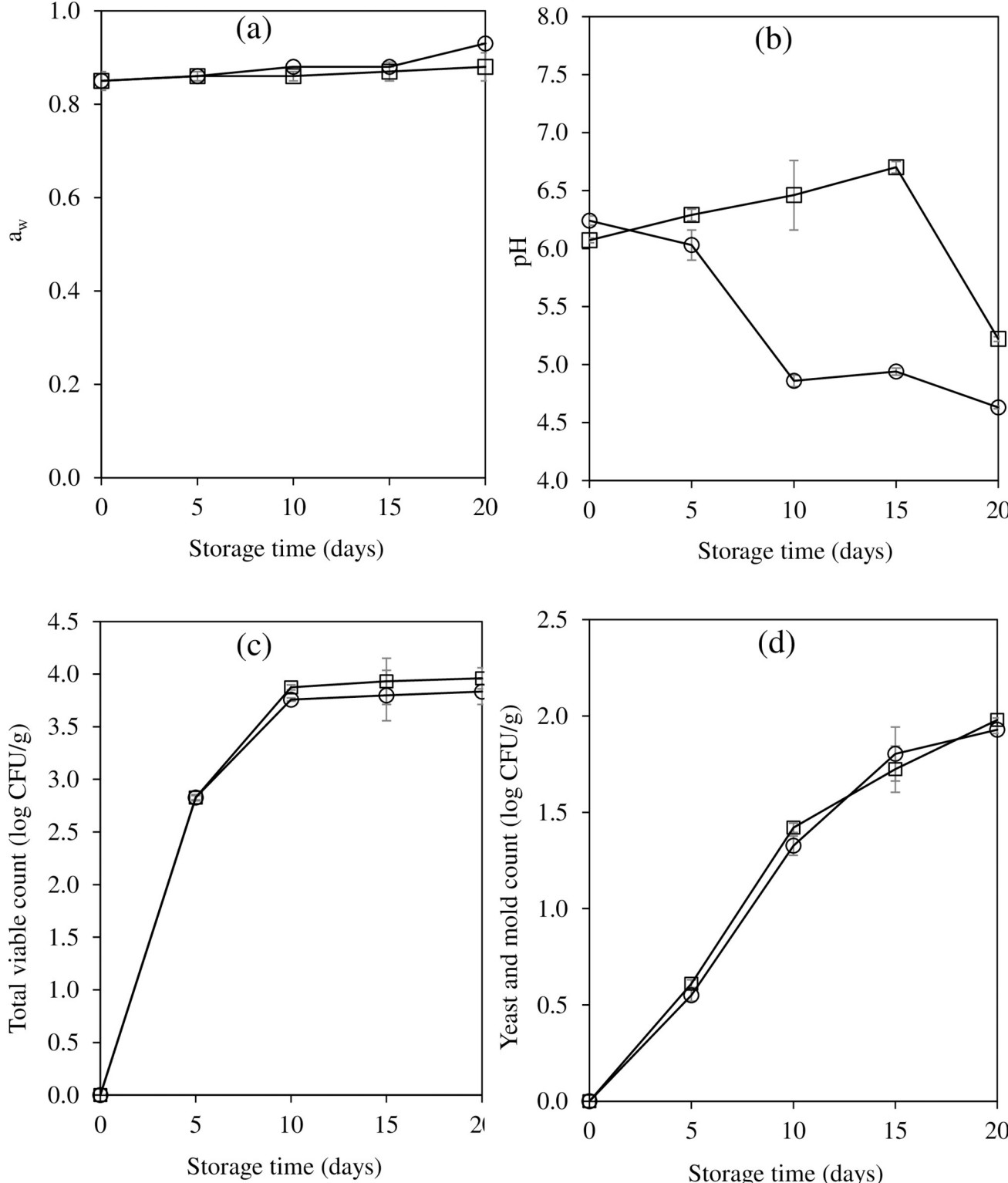

**Fig 4.** Changes in $a_w$ (a), pH (b), total viable count (TVC) (c), and yeast and mold count (d) of semi-dried Chinese style tilapia sausages formulated with pork back fat (control; ○) and carrageenan based emulgel (□) during storage at room temperature.

## Changes in $a_w$, pH, and microbial quality

The $a_w$ of CFS added with EG was remained constant throughout the storage periods (p>0.05) whereas the values of the control increased (p<0.05) (Fig 4A). Results indicated that the EG based CFS can potentially bind water to its matrix when compared to lard based CFS. Carrageenan has an excellent water binding capacity which can reduce the free water in food system [48]. The pH of CFS added with EG slightly increased up to 15 days of storage and then suddenly dropped at Day 20 (Fig 4B). This was might be due to the accumulation of the basic compounds, e.g. ammonia, trimethylamine, and other amine substances, produced by microorganisms and chemical reactions at Day 0–15 whereas the lactic acid bacteria (LAB) might be predominant at the end of storage. LAB contributed to a lower pH by producing lactic acid. Conversely, the pH of the control sausages progressively decreased with time (p<0.05) (Fig 4B), which was possibly due to the noteworthy growth of LAB.

For the microbial quality, no significant differences in TVC (Fig 4C) and yeast and mold counts (Fig 4D) in both samples were observed during storage (p>0.05). This was in agreement with Muguerza et al. [49] and Ruiz-Capillas et al. [50] who found no variations in microbial growth in fermented sausages after replacing pork back fat with oil in water emulsion during storage. During storage, the progressive increases in all TVC and yeast and mold counts were detected at the first 10 days, and then the numbers slightly increased till the end. The growth of microorganisms, mainly LAB, might contribute to a falling in pH (Fig 4B). The CFS was rich in sugar which can be used as a carbon source for LAB fermentation. At the end, the TVC of CFS added with EG and the control reached the maximum value of 3.96 and 3.80 log CFU/g, respectively (Fig 4C), whereas the levels of yeast and mold counts were 1.97 and 1.92 log CFU/g, respectively (Fig 4D). All these values were within the Thai Agricultural Standard (TAS) required for safety (TVC < $10^4$ CFU/g, yeast and mold < $10^2$ CFU/g) [51].

## Conclusion

The incorporation of vegetable oils instead of animal fats can improve the nutritional value of meat products which is one of the alternate choices for health-conscious consumers. EG technology as the modern gastronomy can be applied for this purpose where the oil was filled in the gel matrix based emulsion. This study evaluated the possible effect RBO based EG as a substitute of pork back fat which was used as an ingredient of CFS based on the qualitative examination. Different types of EG rendered the CFS with different physical, chemical, and microstructural characteristics. Overall, carrageenan-EG substitution for pork back fat had negligible impact on chemical composition, textural and microstructural properties of CFS. Carrageenan-EG rendered the CFS with a superior oxidative stability, color retention, and water holing capacity compared to the control during storage at room temperature for 20 days. However, a similar microbial quality was observed among carrageenan-EG formulated CFS and the control. Therefore, carrageenan-EG was a promising lipid carrier for the production of healthier CFS.

## Acknowledgments

We would like to thank Food Technology and Innovation Center of Excellence, Walailak University for providing the scientific and technological equipment for this research.

## Author Contributions

**Conceptualization:** Manat Chaijan, Ling-Zhi Cheong, Worawan Panpipat.

**Data curation:** Manat Chaijan, Worawan Panpipat.

**Formal analysis:** Ling-Zhi Cheong.

**Funding acquisition:** Worawan Panpipat.

**Investigation:** Manat Chaijan, Worawan Panpipat.

**Methodology:** Manat Chaijan, Ling-Zhi Cheong, Worawan Panpipat.

**Writing – original draft:** Manat Chaijan, Ling-Zhi Cheong, Worawan Panpipat.

**Writing – review & editing:** Manat Chaijan, Ling-Zhi Cheong, Worawan Panpipat.

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
