## [Decision Letter · Decision Letter 0]

3 Mar 2021

PONE-D-21-03914

Rice bran oil emulgel as a pork back fat alternate for semi-dried fish sausage

PLOS ONE

Dear Dr. Panpipat,

Thank you for submitting your manuscript to PLOS ONE. After careful consideration, we feel that it has merit but does not fully meet PLOS ONE’s publication criteria as it currently stands. Therefore, we invite you to submit a revised version of the manuscript that addresses the points raised during the review process.

We look forward to receiving your revised manuscript.

Kind regards,

Umakanta Sarker

Academic Editor

PLOS ONE

Journal Requirements:

Reviewers' comments:

Reviewer's Responses to Questions

**Comments to the Author**

1. Is the manuscript technically sound, and do the data support the conclusions?

Reviewer #1: Partly

Reviewer #2: Yes

2. Has the statistical analysis been performed appropriately and rigorously? 

Reviewer #1: Yes

Reviewer #2: Yes

3. Have the authors made all data underlying the findings in their manuscript fully available?

Reviewer #1: Yes

Reviewer #2: No

4. Is the manuscript presented in an intelligible fashion and written in standard English?

Reviewer #1: Yes

Reviewer #2: Yes

5. Review Comments to the Author

Reviewer #1: This paper (PONE-D-21-03914) investigates rice bran oil emulgel as a pork back fat alternate fo 1 r semi-dried fish sausage. The paper contains good information. some revisions are needed for improving the readability of the manuscript.

Some remarks and revisions are listed as follows:

_The abstract should state briefly the purpose of the research, the principal results and major conclusions. A concise and factual abstract is required. Moreover, numerical data of study should be added in abstract section according to the original findings of the study.

_Provide the basic compositions of konjac flour

_Explain the reasons of these changes in color, hardness, and total expressible drip during storage

_In "Results and discussion" section, add more in-depth discussion to explain the differences among samples or during storage, give reasons for the observed phenomena， and compare with previous studies.

_The manuscript to be checked carefully and the typos to be corrected.

_Figure 1 should be updated.  It is not easy to read

Line 73: check the reference style

Line 107: define “gently stirring”

Line 131: what is the selection basis of lipid content of 6.28%?

Line 137: drying time?

Line 144-145: demonstrate the calculation equation of carbohydrate content

Line 148: provide commodity information of pH meter or other setups throughout manuscript

Line 160-161:What is the light source of the color meter? How to test color using colorimeter?

Line 168: what is meaning of thickness of 2.5 cm ? How many pieces were used for textural analysis?

Line 182: give the characteristic information (sex, age, etc.) of planelist, and detailedly describe this content.

Line 204-205: explain the reason of the lowest ash content for Konjac based EG?

Line 213-215: explain the difference of fiber contents

Line 230: check the reference style

Line 233: why does the samples for sodium caseinate based EG have the highest total expressible drip?

Line 259-260: give evidence or data to support this statement.

Line 268: define “MDA”

Reviewer #2: Rice bran oil emulgel as a pork back fat alternate for semi-dried fish sausage

PONE-D-21-03914

The manuscript shows emulgels alternatives for replace lard fat in semi-dried fish sausage. Different structuration components as carrageen, konjac, WPI, and caseinate have been used to structure the water phase. The manuscript is well designed, performed and written. Nevertheless, there are some important points that need clarification before publication. First and more important that is no result that confirms how the phases are distributed, and how the emulsion was formed. For example, all gelators used are hydrophilic how they encapsulate the oil? How can Authors be sure there is an encapsulation? Where is the mechanism discussion? This result need to be added and clarified. Because there is no confirmation that this is an emulgel for example and not a normal oil in water emulsion. Moreover, I have added some small comments to be better explored.

Line 73- Check reference style

Lines 77-79: Why EG is a better option?

Lines 83-84: Can Authors name a few references?

Line 98-128: All over this topic Authors have described a homogenization step but how these several homogenizations were made was never mentioned. Which was the equipment used?

Line 133: How much lard or EG in g was added is missing. All others ingredients were described.

Line 137-139: There is a dry process described in the sausage preparation, but no cooking step. Authors have used cooking in many parts of the manuscript what is wrongly. Please verified if there is any missing information or fixed the nomenclature to meet what was really done.

Line 143: proximate?

Line 162-166: Neither lipid oxidation was based in standard methodologies as AOCS or AOAC, so please describe them.

Line 170-171: English is confusing

Line 172: SEM methodology also needs more description.

Line 178: Why 20 days? Is this the product shelf life?

Line 182: again please describe the methodology appropriated.

Line 189: Manuscript is missing description of replicates, how many process replicates were performed and how many analysis replicates were done?

Line 193: proximate is not an appropriate term

Line 203: This is not true according data on table 1.

Line 230: Check citation style

Line 230-232: This might be a problem in real cooked products not the case.

Line 235-236: It is not clear what Authors mean in this phrase. What are you referring to when you describe muscles proteins?

Table 1: please revise the stats. There is a mistake on PV.

Line 251: I don’t think Authors can affirm this. They have not showed lipid droplet size of the samples to be sure. I think they have failed to explain color differences to be honest.

Line 274: there is no profile in Table 2.

Lines 279- 281: not absolutely true. Authors can’t generalize.

Why measure TPA and all those parameters and not discuss their importance?

Line 326: again oil droplet size was not measure. By the way nothing to assure that there are oils droplets was done.

Lines 329-331: If this is true why the increase was not observed already after 10d ays?

Line 346: Besides the term cooking, encapsulated also need to be carefully use. Where is the result that showed encapsulation? And Authors have used a lot this.

Line 357: Why control sample that have a high amount of saturated fatty acids have showed higher oxidation? There was no discussion on this point.

6. PLOS authors have the option to publish the peer review history of their article (what does this mean?). If published, this will include your full peer review and any attached files.

Reviewer #1: No

Reviewer #2: No

---

## [Author Response · Author response to Decision Letter 0]

17 Mar 2021

Response to Reviewers

All points raised by the reviewers were carefully addressed and answered point-by-point. A revision was made in highlighted red fonts. The revised manuscript was carefully prepared to meet PLOS ONE's style requirements.

Journal Requirements:

Ans: The revised manuscript was carefully prepared to meet PLOS ONE's style requirements.

Reviewer #1: This paper (PONE-D-21-03914) investigates rice bran oil emulgel as a pork back fat alternate fo 1 r semi-dried fish sausage. The paper contains good information. some revisions are needed for improving the readability of the manuscript.

Some remarks and revisions are listed as follows:

_The abstract should state briefly the purpose of the research, the principal results and major conclusions. A concise and factual abstract is required. Moreover, numerical data of study should be added in abstract section according to the original findings of the study.

Ans: Abstract was revised as suggested.

_Provide the basic compositions of konjac flour

Ans: The basic composition of konjac flour was given. “From the technical data sheet, the konjac flour composed of carbohydrate (75%), moisture (12%), protein (8%), fat (<1%), and ash (4%).”

_Explain the reasons of these changes in color, hardness, and total expressible drip during storage

Ans: We tried our best to explain the reasons of these changes in color, hardness, and total expressible drip during storage. Possible mechanisms regarding those changes were given.

_In "Results and discussion" section, add more in-depth discussion to explain the differences among samples or during storage, give reasons for the observed phenomena， and compare with previous studies.

Ans: We tried our best to discuss based on the findings. Some possible mechanisms and supported references were given in the entire manuscript. A revision was made as recommended by the reviewers.

_The manuscript to be checked carefully and the typos to be corrected.

Ans: The entire manuscript was carefully rechecked and all the typo errors were corrected.

_Figure 1 should be updated. It is not easy to read

Ans: All the Figures passed the Preflight Analysis and Conversion Engine (PACE) digital diagnostic tool and met the PLOS requirements.

Line 73: check the reference style

Ans: Done.

Line 107: define “gently stirring”

Ans: Done.

Line 131: what is the selection basis of lipid content of 6.28%?

Ans: This was according to the lipid content in the commercial CFS in Thailand. We have stated this in the Method. “CFS were produced according to the method of Panpipat and Chaijan [12]. The lipid content of all formulae was equally adjusted to 6.28%, according to the lipid content in the commercial CFS in Thailand.”

Line 137: drying time?

Ans: The drying time was given.

Line 144-145: demonstrate the calculation equation of carbohydrate content

Ans: The calculation equation of carbohydrate content was given.

Line 148: provide commodity information of pH meter or other setups throughout manuscript

Ans: The information about the pH meter and other equipment e.g. homogenizer, centrifugation were given.

Line 160-161:What is the light source of the color meter? How to test color using colorimeter?

Ans: The detail is given “(10°standard observers, illuminant D65, Hunter Assoc. Laboratory; VA, USA)”

Line 168: what is meaning of thickness of 2.5 cm ? How many pieces were used for textural analysis?

Ans: Three pieces of CFS with the thickness of 2.5 cm were subjected to texture profile analysis (TPA).

Line 182: give the characteristic information (sex, age, etc.) of planelist, and detailedly describe this content.

Ans: It was our mistake. We have deleted this section because the sensory analysis was not included in this study.

Line 204-205: explain the reason of the lowest ash content for Konjac based EG?

Ans: The possible reason was given. “The lowest ash content in the konjac-EG formulated CFS was possibly due to a low content of ash in the konjac flour.”

Line 213-215: explain the difference of fiber contents

Ans: This was possibly due to different fiber contents in the ingredients.

Line 230: check the reference style

Ans: Done.

Line 233: why does the samples for sodium caseinate based EG have the highest total expressible drip?

Ans: The highest total expressible drip was found in CS-EG made CFS with the pH of 5.5 (Table 1). At a pH of 5.5, which was closer to the muscle protein isoelectric point, the water binding capacity was minimal.

Line 259-260: give evidence or data to support this statement.

Ans: To avoid the over statement, this assumption “Lipid may be tightly packed in the matrices of carrageenan and konjac gels which can prevent the contact with molecular oxygen.” was deleted. 

Line 268: define “MDA”

Ans: The MDA was defined at the first usage already.

Reviewer #2: Rice bran oil emulgel as a pork back fat alternate for semi-dried fish sausage

PONE-D-21-03914

The manuscript shows emulgels alternatives for replace lard fat in semi-dried fish sausage. Different structuration components as carrageen, konjac, WPI, and caseinate have been used to structure the water phase. The manuscript is well designed, performed and written. Nevertheless, there are some important points that need clarification before publication. First and more important that is no result that confirms how the phases are distributed, and how the emulsion was formed. For example, all gelators used are hydrophilic how they encapsulate the oil? How can Authors be sure there is an encapsulation? Where is the mechanism discussion? This result need to be added and clarified. Because there is no confirmation that this is an emulgel for example and not a normal oil in water emulsion.

Ans: As stated in the Introduction, EG is the soft semi-solid materials which is structurally incorporated emulsion droplets into a polymeric gel matrix [14]. From the principle, the gelators did not encapsulate the oil directly but the emulsified oil was entrapped in the gel structure. Since the emulsion was firstly prepared, followed by the incorporation of the emulsion into the gel. As you can see the pictures in Fig.1, the EG were formed in all cases.

Moreover, I have added some small comments to be better explored.

Line 73- Check reference style

Ans: Done.

Lines 77-79: Why EG is a better option?

Ans: EG could be a better option in comminuted emulsion like sausage than simple emulsified oil to achieve a better physical characteristic close to animal fat with greater physical stability because the product stability can greatly be improved by immobilizing oil droplets within a three dimensional gel network, resulting in the prevention of flocculation and coalescence [14]. A better texture, higher water holding capacity, and lower cooking loss of products containing EG can be obtained compared to simple emulsified oil [15].

Lines 83-84: Can Authors name a few references?

Ans: Done.

Line 98-128: All over this topic Authors have described a homogenization step but how these several homogenizations were made was never mentioned. Which was the equipment used?

Ans: The equipment was given. 

Line 133: How much lard or EG in g was added is missing. All others ingredients were described.

Ans: Lard or EG were added with different contents depending on their fat contents in order to obtain a similar fat level of 6.28%. We originally stated in the Method. “CFS were produced according to the method of Panpipat and Chaijan [12]. The lipid content of all formulae was equally adjusted to 6.28%, according to the lipid content in the commercial CFS in Thailand. To obtain a similar fat level, each EG was calculated based on lipid content and used to replace the pork back fat. Briefly, tilapia mince (62.8 g) was handed mixed with lard or EG (to obtain 6.28% fat), sugar (15.7 g), sodium nitrite (1.73 g), and sodium erythorbate (0.15 g) for 8 min.”

Line 137-139: There is a dry process described in the sausage preparation, but no cooking step. Authors have used cooking in many parts of the manuscript what is wrongly. Please verified if there is any missing information or fixed the nomenclature to meet what was really done.

Ans: As stated in the Introduction, this kind of sausage is a semi-dried sausage. “Chinese-style fish sausage (CFS) or Pla Chiang (in Thai) is a semi-dried coarse sweet sausage produced by hand kneading coarse ground fish and coarse ground animal fat with salt, nitrite, spices, and sugar. After stuffing in the edible casing, the drying is performed to lower the moisture content to 25% for extending the shelf life [12].” To clarify the way of cooking, a sentence was added in the Introduction “CFS is needed to be cooked e.g. pan frying before consumption.”. The term “cooking” was avoid in this manuscript.

Line 143: proximate?

Ans: Yes it is. Proximate analysis is used to estimate the amounts of moisture, protein, lipid, ash, fiber, and carbohydrate in any sample using the standard method of AOAC.

Line 162-166: Neither lipid oxidation was based in standard methodologies as AOCS or AOAC, so please describe them.

Ans: The methods for PV and TBARS determinations were detailed.

Line 170-171: English is confusing

Ans: The sentence was rewritten. 

Line 172: SEM methodology also needs more description.

Ans: The SEM methodology was given in detail.

Line 178: Why 20 days? Is this the product shelf life?

Ans: The shelf life of this product is about 15-20 days. So, in this study, a test was conducted up to 20 days.

Line 182: again please describe the methodology appropriated.

Ans: It was our mistake. We have deleted this Section because the sensory analysis was not included in this study.

Line 189: Manuscript is missing description of replicates, how many process replicates were performed and how many analysis replicates were done?

Ans: A completely randomized design was used in this study and the entire experiment was replicated three times. The triplicate determinations were done for all analyses.

Line 193: proximate is not an appropriate term

Ans: We would like to keep this term because we did the proximate analysis.

Line 203: This is not true according data on table 1.

Ans: Thank you very much. It was rewritten to comply with the data on Table 1.

Line 230: Check citation style

Ans: Done.

Line 230-232: This might be a problem in real cooked products not the case.

Ans: Thank you for your suggestion. However, we would like to keep this assumption because heat treatment was applied during drying.

Line 235-236: It is not clear what Authors mean in this phrase. What are you referring to when you describe muscles proteins?

Ans: The term “muscle” was deleted. In this sentence, we would like to find the statement to support the drip. “Results implied that the proteins did not effectively bind water in the sausage matrix when EG was added. Overall, all EG might partially destabilize the sausage structure, resulting in the imbalance among protein-protein, protein-water, and protein-fat interactions”.

Table 1: please revise the stats. There is a mistake on PV.

Ans: Thank you very much. It was a typo error. It was corrected accordingly.

Line 251: I don’t think Authors can affirm this. They have not showed lipid droplet size of the samples to be sure. I think they have failed to explain color differences to be honest.

Ans: We deleted this assumption to avoid the over statement. 

Line 274: there is no profile in Table 2.

Ans: It was changed to “textural properties”.

Lines 279- 281: not absolutely true. Authors can’t generalize.

Ans: We reported this due to the statistical results. “Compared to the control, CFS added with carrageen-EG and WPI-EG tended to have comparable hardness, springiness, adhesiveness, gumminess, and chewiness to the control (p>0.05).”

Why measure TPA and all those parameters and not discuss their importance?

Ans: We tried our best to discuss the textural properties of CFS and the main finding was that the textural parameters of CFS were governed by EG type. All TPA terms including hardness, springiness, cohesiveness, adhesiveness, gumminess, and chewiness were mentioned. The possible links between the characteristics of EG, changes in proteins, and the textural properties of the products were discussed. 

Line 326: again oil droplet size was not measure. By the way nothing to assure that there are oils droplets was done.

Ans: We deleted this assumption to avoid the over statement.

Lines 329-331: If this is true why the increase was not observed already after 10 days?

Ans: From the results, the hardness of EG formulated CFS remained constant over the storage period (p>0.05), while an increasing trend in hardness was found in the control (p<0.05) (Fig. 2d). So, we stated that “Results indicated that the EG was able to maintain the hardness of the sausage during storage. An increase in hardness value of the control sample was probably due to the water loss during storage (Fig. 2e).”

Line 346: Besides the term cooking, encapsulated also need to be carefully use. Where is the result that showed encapsulation? And Authors have used a lot this.

Ans: Actually, “encapsulation” can be defined as a process for entrapping one substance within another substance. In the case of emulgel, emulsified lipid was entrapped in the structure of gelling agent. However, the term “incorporation” was used instead of “encapsulation” in this context.

Line 357: Why control sample that have a high amount of saturated fatty acids have showed higher oxidation? There was no discussion on this point.

Ans: The discussion on this point was given. “The results indicated that an incorporation of RBO in carrageenan gel matrix prior to use in CFS production could potentially protect the autooxidation. de Souza Paglarini et al. [36] reported that TBARS of soybean oil EG formulated Bologna sausage increased during storage but the contents were below the rancidity limits. Although lard contains higher amount of saturated fatty acids, it still contains unsaturated fatty acids which can be oxidized. In addition, the EG structure may have helped to prevent the susceptibility of lipid oxidation. In EG formulated sausage, oils might be buried in the gel matrix which can restrict the oxygen penetration. Thus, the lipid oxidation in control was higher than sample with EG.” 

Ans: Done.

---

## [Decision Letter · Decision Letter 1]

30 Mar 2021

PONE-D-21-03914R1

Rice bran oil emulgel as a pork back fat alternate for semi-dried fish sausage

PLOS ONE

Dear Dr. Panpipat,

Thank you for submitting your manuscript to PLOS ONE. After careful consideration, we feel that it has merit but does not fully meet PLOS ONE’s publication criteria as it currently stands. Therefore, we invite you to submit a revised version of the manuscript that addresses the points raised during the review process.

ACADEMIC EDITOR: According to the comments of reviewer 2 and my opinion as an academic editor, again the authors are suggested to revise the manuscript to improve the missing discussion highlighting the explanation of mechanisms, emulgel structuration, texture, and SEM pictures with available references in literature to support the results.  

We look forward to receiving your revised manuscript.

Kind regards,

Umakanta Sarker

Academic Editor

PLOS ONE

Reviewers' comments:

Reviewer's Responses to Questions

**Comments to the Author**

1. If the authors have adequately addressed your comments raised in a previous round of review and you feel that this manuscript is now acceptable for publication, you may indicate that here to bypass the “Comments to the Author” section, enter your conflict of interest statement in the “Confidential to Editor” section, and submit your "Accept" recommendation.

Reviewer #1: All comments have been addressed

Reviewer #2: (No Response)

2. Is the manuscript technically sound, and do the data support the conclusions?

Reviewer #1: Yes

Reviewer #2: Partly

3. Has the statistical analysis been performed appropriately and rigorously? 

Reviewer #1: Yes

Reviewer #2: Yes

4. Have the authors made all data underlying the findings in their manuscript fully available?

Reviewer #1: (No Response)

Reviewer #2: No

5. Is the manuscript presented in an intelligible fashion and written in standard English?

Reviewer #1: Yes

Reviewer #2: Yes

6. Review Comments to the Author

Reviewer #1: I have no further comments therefore I suggest that the manuscript can be accepted in its present form

Reviewer #2: Authors have addressed all minor comments I have made in previously version. Nevertheless, the major comment regarding the mechanism and lack of explanation on the emulgel structuration is a issue that was not answered. Although authors have said they have done their best to explain texture and SEM pictures for both reviewers questions, however they have not even tried to improve. There are a lot of references available to support these missing discussion. I believe only shows the changing in physical properties is not enough for a scientific paper. The explanations of all whys have always to followed the changes.

Moreover, authors forgot to highlight the changes as they have said they did. What makes really hard to follow the changes.

7. PLOS authors have the option to publish the peer review history of their article (what does this mean?). If published, this will include your full peer review and any attached files.

Reviewer #1: No

Reviewer #2: No

---

## [Author Response · Author response to Decision Letter 1]

2 Apr 2021

Response to Reviewers

All points raised by the editor and reviewer were carefully addressed and answered point-by-point. A revision was made in highlighted red fonts. The revised manuscript was carefully prepared to meet PLOS ONE's style requirements.

ACADEMIC EDITOR: According to the comments of reviewer 2 and my opinion as an academic editor, again the authors are suggested to revise the manuscript to improve the missing discussion highlighting the explanation of mechanisms, emulgel structuration, texture, and SEM pictures with available references in literature to support the results. 

Ans: The manuscript was revised accordingly in which the mechanisms, emulgel structuration, texture, and SEM pictures were discussed. 

Reviewer #2: Authors have addressed all minor comments I have made in previously version. Nevertheless, the major comment regarding the mechanism and lack of explanation on the emulgel structuration is a issue that was not answered. Although authors have said they have done their best to explain texture and SEM pictures for both reviewers questions, however they have not even tried to improve. There are a lot of references available to support these missing discussion. I believe only shows the changing in physical properties is not enough for a scientific paper. The explanations of all whys have always to followed the changes.

Ans: Thank you very much for your invaluable suggestions. A revision was made carefully as suggested. The emulgel structuration, texture, and SEM pictures were discussed intensively (See L293-L262). The references were also updated.

1. Gani A, Benjakul S. Impact of virgin coconut oil nanoemulsion on properties of croaker surimi gel. Food Hydrocoll. 2018;82:34-44.

2. Chandra MV, Shamasundar BA. Texture profile analysis and functional properties of gelatin from the skin of three species of fresh water fish. Int. J. Food Prop. 2015;18:572-584.

3. Ashara KC, Paun JS, Soniwala MM, Chavada JR, Mori NM. Micro-emulsion based emulgel: a novel topical drug delivery system. Asian Pac. J. Trop. Dis. 2014;4:S27-S32.

4. Cubides YTP, Eklund PR, Foegeding EA. Casein as a modifier of whey protein isolate gel: sensory, texture and rheological properties. J. Food Sci. 2019;84:3399-3410.

5. Zhang T, Xu X, Ji L, Li Z, Wang Y, Xue Y, Xue C. Phase behaviors involved in surimi gel system: Effects of phase separation on gelation of myofibrillar protein and kappa-carrageenan. Food Res. Int. 2017;100:361-368.

6. Ayadi MA, Kechaou A, Makni I, Attia H. Influence of carrageenan addition on turkey meat sausages properties. J. Food Eng. 2009;93:278-283.

7. Sun XD, Holley RA. Factors influencing gel formation by myofibrillar proteins in muscle foods. Compr. Rev. Food Sci. Food Saf. 2011;10:33-51.

8. Liu J, Zhu K, Ye T, Wan S, Wang Y, Wang D, Wang C. Influence of konjac glucomannan on gelling properties and water state in egg white protein gel. Food Res. Int. 2013;51:437-443.

9. Poklukar K, Čandek-Potokar M, Batorek Lukač N, Tomažin U, Škrlep M. Lipid deposition and metabolism in local and modern pig breeds: A review. Animals. 2020;10:424.

10. de Figueiredo Furtado G, Mantovani RA, Consoli L, Hubinger MD, da Cunha RL. Structural and emulsifying properties of sodium caseinate and lactoferrin influenced by ultrasound process. Food Hydrocoll. 2017;63:178-188.

11. Leon AM, Medina WT, Park DJ, Aguilera JM. Properties of microparticles from a whey protein isolate/alginate emulsion gel. Food Sci. Technol. Int. 2018;24:414-423.

12. Ortiz J, Aguilera JM. Effect of kappa-carrageenan on the gelation of horse mackerel (T. murphyi) raw paste surimi-type. Food Sci. Technol. Int. 2004;10:223-232.

Moreover, authors forgot to highlight the changes as they have said they did. What makes really hard to follow the changes.

Ans: In the revision 1, there were 2 files including ‘manuscript’ and ‘manuscript with track changes’. We highlighted what we have changed in the red fonts. In this revision (R2), the revision was made in the highlighted red fonts as well.

---

## [Decision Letter · Decision Letter 2]

8 Apr 2021

Rice bran oil emulgel as a pork back fat alternate for semi-dried fish sausage

PONE-D-21-03914R2

Dear Dr. Panpipat,

We’re pleased to inform you that your manuscript has been judged scientifically suitable for publication and will be formally accepted for publication once it meets all outstanding technical requirements.

Kind regards,

Umakanta Sarker

Academic Editor

PLOS ONE

Additional Editor Comments (optional):

Reviewers' comments:

Reviewer's Responses to Questions

**Comments to the Author**

1. If the authors have adequately addressed your comments raised in a previous round of review and you feel that this manuscript is now acceptable for publication, you may indicate that here to bypass the “Comments to the Author” section, enter your conflict of interest statement in the “Confidential to Editor” section, and submit your "Accept" recommendation.

Reviewer #2: All comments have been addressed

2. Is the manuscript technically sound, and do the data support the conclusions?

Reviewer #2: Yes

3. Has the statistical analysis been performed appropriately and rigorously? 

Reviewer #2: Yes

4. Have the authors made all data underlying the findings in their manuscript fully available?

Reviewer #2: No

5. Is the manuscript presented in an intelligible fashion and written in standard English?

Reviewer #2: Yes

6. Review Comments to the Author

Reviewer #2: No further comments, major comments have been addressed. I believe the manuscript is suitable for publication, good job.

7. PLOS authors have the option to publish the peer review history of their article (what does this mean?). If published, this will include your full peer review and any attached files.

Reviewer #2: No

---

## [Editor Report · Acceptance letter]

13 Apr 2021

PONE-D-21-03914R2 

Rice bran oil emulgel as a pork back fat alternate for semi-dried fish sausage 

Dear Dr. Panpipat:

I'm pleased to inform you that your manuscript has been deemed suitable for publication in PLOS ONE. Congratulations! Your manuscript is now with our production department. 

Kind regards, 

on behalf of

Professor Umakanta Sarker 

Academic Editor

PLOS ONE